Low-diameter topic-based pub/sub overlay network construction with minimum–maximum node degree

Yumusak Semih semih.yumusak@karatay.edu.tr 1
Layazali Sina 2
Oztoprak Kasim 3
Hassanpour Reza 4
1 Department of Computer Engineering, KTO Karatay University , Konya , Turkey
2 Department of Computer Engineering, Çankaya University , Ankara , Turkey
3 Department of Computer Engineering, Konya Food and Agriculture University , Konya , Turkey
4 Department of Computer Science, Rotterdam University of Applied Sciences , Rotterdam , Holland
Strufe Thorsten
Electronic publication date: 2021 May 14
Publication date: 2021
Volume: 7
Electronic Location ID: e538
Received 2020 Oct 22; Accepted 2021 Apr 20
Copyright: ©2021 Yumusak et al.
Copyright year: 2021
Copyright holder: Yumusak et al.
License: This is an open access article distributed under the terms of the Creative Commons Attribution License, which permits unrestricted use, distribution, reproduction and adaptation in any medium and for any purpose provided that it is properly attributed. For attribution, the original author(s), title, publication source (PeerJ Computer Science) and either DOI or URL of the article must be cited.
License URL: https://creativecommons.org/licenses/by/4.0/

Keywords: Overlay network design, Peer-to-peer networks, Publisher/subscriber systems

Funding: KTO Karatay University This work was supported by KTO Karatay University. There was no additional external funding received for this study. The funders had no role in study design, data collection and analysis, decision to publish, or preparation of the manuscript.

==============================
In the construction of effective and scalable overlay networks, publish/subscribe (pub/sub) network designers prefer to keep the diameter and maximum node degree of the network low. However, existing algorithms are not capable of simultaneously decreasing the maximum node degree and the network diameter. To address this issue in an overlay network with various topics, we present herein a heuristic algorithm, called the constant-diameter minimum–maximum degree (CD-MAX), which decreases the maximum node degree and maintains the diameter of the overlay network at two as the highest. The proposed algorithm based on the greedy merge algorithm selects the node with the minimum number of neighbors. The output of the CD-MAX algorithm is enhanced by applying a refinement stage through the CD-MAX-Ref algorithm, which further improves the maximum node degrees. The numerical results of the algorithm simulation indicate that the CD-MAX and CD-MAX-Ref algorithms improve the maximum node-degree by up to 64% and run up to four times faster than similar algorithms.

Introduction

In publish/subscribe (pub/sub) systems, publishers forward different types of messages to specific subscribers in a decoupled mode. Publishers broadcast information through logical channels, while subscribers receive them based on their topic interests. Pub/sub systems are divided into two different categories, namely topic- and content-based categories.

In topic-based pub/sub systems, publishers broadcast their messages based on the topic of the message. Each topic exclusively pertains to a specific logical channel. Subsequently, as stated in Oztoprak & Akar (2007), subscribers receive all messages associated with the topics to which they have subscribed to. Consequently, as stated in Milo, Zur & Verbin (2007), all messages about those topics will be sent to every user who has joined that particular group. Publishers take the responsibility of classifying the messages that subscribers receive.

In contrast, in content-based pub/sub systems, subscribers only receive messages whose attributes match with their interests; hence, Carvalho, Araujo & Rodrigues (2005) states that these attributes characterize the logical channels . In this category, the matching algorithm between the publishers and the subscribers is based on the attribute values referred to as the content. In other words, the receivers decide which messages they would receive. In both cases, however, publish/subscribe systems show similarities with Information Centric Networks in terms of sharing/distributing information among users.

Pub/sub systems have a variety of use cases. As explained by O’Riordan (2021), a simple use case of a pub/sub system may be a chat application where a participant can subscribe to any of the chat rooms which has a designated pub/sub topic. When a user sends a message to a chat room, the message is published on that topic of the chat room. The subscribers of the topic/chat room receive the message. As stated by Google Cloud (2021), the pub/sub systems fit best, when there is a need for durable message storage and real-time delivery for those messages with high availability for massive scale. These features are the foundation of cloud computing where pub/sub systems are heavily used. Balancing load among network clusters, implementation of asynchronous workflows, distributing event notifications, data streaming from various processes or devices are examples to pub/sub systems. Apache (Kafka, 2021), Microsoft Azure (2021), Google cloud (2021) and Amazon Web Services (AWS) (2021) are examples of popular pub/sub systems.

For each topic t ∈ T in a typical fully decentralized topic-based pub/sub system based on the peer-to-peer (P2P) connected overlay, a sub graph is derived using the nodes interested in t. Hence, the nodes interested in topic t do not need to rely on other nodes to send or receive their messages. Every node must maintain its connections (e.g., checking the accessibility of neighbors) and monitor data streaming through the connections; therefore, overlay networks with a low maximum number of links emitting from a node and low network diameters are desirable. If a proper correlation exists between node subscriptions, the connectivity of many topics subscribed by those two nodes will be satisfied by adding only one edge between the two nodes. Hence, the maximum number of links of a node and the total number of overlay connections will considerably diminish. The significance and the impact of the topic correlation of nodes in optimizing the overlay networks were highlighted in a relevant paper by Chockler et al. (2007a). A constructed sub-graph acts as a selective message delivery among different publishers and subscribers belonging in the same interest group. In a sub-graph, messages are routed to the node destination (subscriber) with the same topic interest. Generally, a sub-group of pub/sub system can be modelled as a trimerous <Π, β, Σ > collections of functions. The sets involved are determined based on their functionality: Π = p0, …, pi − 1 is a set of i processes in the system that act as publishers providing information to those needing it. Σ = C0, …, Cj − 1 is a set of j processes referred to as subscribers that are known as consumers of information provided by publishers. The set of publishers and the set of subscribers can have non-zero intersection, which means that the process can also operate as a publisher and a subscriber at the same time (decoupling). Pub-sub systems are decoupled; therefore, a process may change anything about a publisher if it does not change the way it produces messages. Hence, there is no need to change something about the downstream subscribers. Similarly, the opposite process is true as well. Systems with decoupling mechanisms do not need to consider any issues such as addressing and synchronization. β = B0, …, Bk − 1 presents a logical centralized service that enables publishers and subscribers to connect. In other words, any publisher or subscriber in a sub group may exclusively send or receive specific information through β. Not only does β provide communication between publishers and subscribers, publishers and subscribers are kept in a decoupled mode during the communication process.

Baldoni et al. (2007) and Lau et al. (2009) states that, reducing the maximum number of links of an overlay can play a vital role in various network fields, such as survivable and wireless network design. Chockler et al. (2007a) presented the concept of topic connectivity, in which an individual overlay network connects nodes with similar topics. They introduced the Greedy Merge (GM) algorithm to construct an overlay with the least possible number of connections. A number of other solutions for the overlay design were also recently introduced by Carvalho, Araujo & Rodrigues (2005). However, all of the existing methods (Chen, Jacobsen & Vitenberg, 2010a; Chen et al., 2015; Chen, Jacobsen & Vitenberg, 2010b) suffer from either a high diameter or a high maximum-node-degree. In addition, all constant-diameter algorithms connect the nodes in a star topology manner, resulting in the best possible diameter while giving rise to nodes with high node degrees. Chockler et al. (2007b) states that, these nodes are responsible for managing a large number of connections to their neighbors, which results in a high traffic overhead.

In this study, we propose an algorithm for constructing a scalable topic-connected overlay (TCO), which has a low maximum node degree and a constant diameter of 2, to solve the above-mentioned problems. The proposed algorithms (i.e., CD-MAX and CD-MAX-Ref) outperform the existing algorithms in the literature in terms of constructing optimum overlays with a minimum node degree and a small diameter. In addition, the performance of the proposed CD-MAX algorithm in terms of the required running time for constructing overlays provides a suitable conformance on scalability requirements.

The remainder of this paper is presented as follows: ‘Overlay Design Algorithms’ provides a summary of the previous studies on pub/sub networks, including the concept of the GM algorithm and other algorithms proposed for building overlays with a minimum number of edges; ‘Proposed Algorithm’ presents the details of the proposed CD-MAX and CD-MAX-Ref algorithms; ‘Experimental Results’ provides the comparative results of the CD-MAX and CD-MAX-Ref algorithms against the most recent methods from the literature; and ‘Conclusions and Future Work’ presents our drawn conclusion and the possible future directions.

Overlay Design Algorithms

An effective publication routing protocol is needed in designing an efficient pub/sub system, and it can play a vital role on the system performance. Therefore, Onus & Richa (2011) stated that the quality of a constructed overlay can be assessed based on the complexity of the routing scheme applied. The complexity can be minimized if all the nodes interested in a topic t ∈ T can be organized into a dissemination tree. In this case, as Chockler et al. (2007a), Chockler et al. (2007b) states, the topic dissemination trees should have the following issues:

• Each tree for topic m includes only the nodes interested in topic m

• The diameter of the topic trees should be low

The GM algorithm by Chockler et al. (2007a) solves the two issues by the low-diameter publish/subscribe overlay algorithms. Suppose that G is an overlay network, and the essential prerequisite to solving issue (1) is to ensure topic connectivity, where a sub-graph connects all the nodes interested in topic m. Chockler et al. (2007a) introduced the topic connectivity concept and the minimum topic-connected overlay problem. They provided an approximation solution, called the GM algorithm, for the problem with the minimum number of links.

The GM algorithm begins with the overlay network G = (N, ∅). There are m ∈ M|n:Int(n, m) = 1| individual topic-connected components of G for each topic m ∈ M.Int(x, m) indicates whether or not node x is interested in topic m. The algorithm continues by connecting two nodes at each repetition until the resulting overlay comprises maximally one topic-connected component for each m ∈ M. The CD-ODA-II algorithm by Onus & Richa (2011) initializes with G = (N, ∅) as the overlay network. A node u which has the maximum connection density is chosen in each iteration. Afterwards, edges are added between u and its neighbors. Thereafter, the interest topics of u are removed from the set of topics. Unlikely, the 2D-ODA algorithm by Onus & Richa (2016) starts with G = (V, E) as the overlay network and a topic set T is selected in which that topic is in the interest of node u and the selected topic has the maximum node density. The node with maximum topic density for a topic is chosen at each iteration, then together with the star structure, node is added to the network. After all, the topic is removed from the set.

Publish/subscribe challenges

The following three main challenges must be handled in building an effective pub/sub system: (1) expression of the interest to the topics by the subscribers, (2) organization of the notification service to deliver interests to topics, and (3) delivery of messages to the subscribers by the publishers. These states are strongly coupled, and their contribution can affect the system performance. For instance, as described in Triantafillou & Aekaterinidis (2004), a rudimentary subscription algorithm may improve the functionality of multicasting, but it facilitates a poor expression ability for subscribers to announce the topics they are interested in. Eugster et al. (2003) states that the architecture of the pub/sub systems can generally be divided into client–server groups and P2P. In client–server architectures, the servers are the providers of information (publishers), while the clients are the subscribers. Intermediate nodes, called brokers, have been introduced to decouple clients and servers and achieve a better performance. Therefore, these architectures are referred to as broker-based architectures. Meanwhile, in P2P architectures, each node performs as either subscribers or publishers or both. In a P2P paradigm, all nodes can operate under various roles (e.g., subscriber, root, or internal node of a multicast tree) and play under a combination of these roles. The P2P architectures are also called decentralized architectures. A typical pub/sub system must hold two principle characteristics of P2P networks: (i) scalability and (ii) fault tolerance/reliability. The following sub-sections briefly introduce both architectures.

Preliminaries

An overlay is defined as an undirected graph G(V,E), where V is the set of nodes, and E is the set of edges. The number of nodes interested in at least one topic, which node u is interested in, is called the node u interest group, which is computed as nu=v∈V∃t∈T,Intv,t=Intu,t=1|

T is the set of topics, which a subscriber can be interested in. Int(u, t) = 1 indicates that subscriber u is interested in topic t, while Int(u, t) = 0 indicates otherwise. The degree of node u denoted by du is defined as the total number of edges incident to it and given as du=v∈Vu,v∈E|

The degree of topic t is defined as the number of subscribers interested in that topic as dt=v∈VIntv,t| The density of node u is given by densityu=∑t∈Tv∈VIntv,t=Intu,t=1|v∈V∃t∈T,Intv,t=Intu,t=1|.

Additionally, the diameter of a graph is the length of the shortest path between the two nodes which are farthest from each other.

Proposed Algorithm

Most of the approaches used for designing scalable overlay networks failed to achieve an appropriate trade-off between the maximum node degree and the diameter of the overlay network. On the one hand, solutions for decreasing the number of connections exist. Chockler et al. presented the problem of constructing overlay networks with the least possible links. They considered this issue as an NP-Complete problem and proposed the GM algorithm to solve it Chockler et al. (2007a). The GM algorithm begins with the overlay network G = (N, ∅). There are ∑m ∈ M|n:Int(n, m) = 1| individual topic-connected components of G for each topic m ∈ M.Int(x, m) indicates whether or not node x is interested in topic m. The algorithm continues by connecting two nodes at each repetition until the resulting overlay comprises maximally one topic-connected component for each m ∈ M. The two nodes connected during each repetition are those with the greatest number of topics in common.

On the other hand, a number of solutions presented in the previous section provide overlays with a low diameter, but the maximum node degree of these overlays is considerably high Carvalho, Araujo & Rodrigues (2005) Onus & Richa (2011). We propose herein an algorithm, called Low-diameter Topic-based Pub/Sub Overlay Network with Minimum–Maximum Node Degree (CD-MAX), to simultaneously address both issues. Our proposed method improves the CD-MAX algorithm based on building an overlay using a star topology to avoid a high node degree. The proposed CD-MAX algorithm (Algorithm 1) creates an initial overlay network by selecting the nodes with a minimum interest group at each iteration.

This initial overlay network is refined to further improve the maximum node degree by the CD-MAX-Ref algorithm (Algorithm 2). A topic-connected overlay is defined as; for each topic t, all possible node n interested in that topic are connected. If there is a node interested in topic t which is not subscribed by other nodes throughout the network, the overlay for that topic is connected. In constructing a network, the proposed CD-MAX algorithm selects the node with the smallest interest group (nu) and connects it to every node in its interest group. If more than two nodes with an equal smallest interest group are found, the algorithm will select the node with the highest connection density. Each selected node takes the responsibility of topics to which it has subscribed to. These topics are removed from the list of topics, and the algorithm iterates until the list of topics becomes empty. Algorithm 1 presents this procedure. In the last step of Algorithm 1, CD-MAX algorithm finds the node with the smallest nu that is interested in the subsets of topics t. The selected node u will be connected to its neighbors and the topics subscribed by node u will be removed from the original topic set. In order to proceed, a proof that the CD-MAX algorithm terminates in O(|V|2∗|T|2) is provided below.

Lemma 1 The running time of CD-MAX algorithm is O(|V|2∗|T|2).

Proof In Algorithm 1, between lines 5 and 24, the outer loop (the while loop on line 5) iterates T times. In addition, the inner loop (the for loop on line 6) iterates V times. In the worst case, calculation of interest group takes V times. Thus, finding a node with minimum node degree takes O(|V|2∗|T|). When node u with minimum node degree has been selected, all the topics to which node u is subscribed is removed. At each iteration, one topic is removed from the original topic set. Thus algorithm takes O(|V|2∗|T|)∗|T| = O(|V|2∗|T|2) time steps to terminate.

Lemma 2 CD-MAX algorithm guarantees the diameter of at most 2 for each topic.

Proof Since CD-MAX diameter provides a star topology for each topic, each node requires at most 2 edges to receive and send any topic it is interested in.

Lemma 3 The space complexity of CD-MAX algorithm is O(|T|∗|V|).

Proof CD-MAX uses a star topology for each topic, each node requires at most 2 edges to receive and send any topic it is interested in. Hence, it gives us a space use of 2∗|T∗V| which gives a space complexity of O(|T|∗|V|).

As a justification of the algorithm; at each iteration, at least one node is connected to its neighbors (one edge is required), meaning that finding a node with minimum node degree is achieved.

After the CD-MAX implementation, a refinement process, called CD-MAX-Ref, is applied to the resulting overlay network to further improve the maximum node degree. CD-MAX-Ref checks if any node with a lower node degree du exists for each topic. The topic overlay center becomes the newly found alternative node if CD-MAX-Ref finds an alternative node. The edges connecting the current center node of the topic with the subscribers to that topic will be removed, and the alternative node is connected to all the subscribers of that topic. Consequently, the new node becomes the center of the topic(s) subscribed by the current node. The center of the overlay topics subscribed by the current node may be at more than one node. CD-MAX-Ref can be used independently with any overlay network from CD-MAX. Algorithm 2 formally describes the CD-MAX-Ref algorithm.

The CD-MAX-Ref algorithm (Algorithm 2) takes over when CD-MAX terminates. It examines all topics and finds the center node for each topic. The center node is a node interested in topic t ∈ T which is selected to connect all nodes interested in topic t ∈ T. It then searches the overlay for an alternative node, which will have a lower node degree if it becomes the center node for that topic. If CD-MAX-Ref manages to locate such a node, the edges corresponding to that topic are removed from the center node of the topic, and the newly discovered nodes are connected to the subscribers of that topic. CD-MAX-Ref can decrease the maximum node degree obtained from the CD-MAX algorithm. To prove the improvement made by the CD-MAX-Ref algorithm, let us assume that u has a node degree of du, and is the center for the k topics. If a node, such as v, is found satisfying dv + dt < du − dt, the algorithm eliminates all the edges connecting u to the other nodes, except v, and adds edges from v to those nodes. If the m edges are eliminated in this stage, the same number of new edges should be added to connect the neighbors of u to v. However, v already has some edges reflected in dv; hence, the probability of having an overlap between the added and existing edges in v exists. In some cases, the number of added edges will be less than the number of removed edges. In the worst case, CD-MAX-Ref will add the same number of edges to v as was deleted from u. Even in this case, the node degree of v will be smaller (according to the assumption), thereby resulting in a lower maximum node degree. Considering that the node with the highest node degree is processed in the same manner, the algorithm will reduce its node degree and the maximum node degree of the network. Additionally, the CD-MAX-Ref algorithm is not affected on how many topics are associated with a removed edge. In any case, CD-MAX-Ref finds an alternative node for each topic that is related to the removed edge. Before moving forward we prove that the CD-MAX-Ref algorithm terminates in O(|T|2∗|N|3).

Lemma 4 The running time of CD-MAX-Ref algorithm is O(|T|2∗|N|3).

Proof CD-MAX-Ref takes over where CD-MAX terminates which runs in O(|T|2∗|N|2). It then examines all topics and finds the center nodes for each topic in O(|N|) in worst case. Thus algorithm takes O(|T|2∗|N|3) time steps to be terminated.

Lemma 5 The space complexity of CD-MAX-Ref algorithm is O(|T|∗|N|).

Proof Since CD-MAX-Ref builds over CD-MAX, it uses similar data structure with more iteration on the same space which does not extend the use of space. Hence, it gives us a space use of in the order of |T∗N| giving a space complexity of O(|T|∗|N|).

The complexity of a constructed pub/sub overlay network can be determined through the cost of broadcast consideration. The total time required to send a chunk of information from a publisher to specific group of nodes that subscribe certain type of information is a crucial factor that truly depends on the resulted diameter between publishers and subscribers. Meanwhile, due to bandwidth and memory constraints, it is also necessary to keep the maximum node degree of an overlay low. For example, a star topology would be the best option in terms of overlay diameter. But, on the other hand, in terms of memory, bandwidth and energy consumption, it will become a big problem in this scenario when number of nodes grows in the network. Hence, providing a trade-off between diameter and maximum node degree of a pub/sub overlay is an ideal way in pub/ sub overlay network design.

Evaluation of CD-MAX with examples

Examples 1 and 2 are presented to clarify the steps used by the proposed CD-MAX and CD-MAX-Ref algorithms. Examples 3–5 additionally compare CD-MAX and CD-MAX-Ref algorithms over other constant diameter (CD) overlay design algorithms (CD-ODA, CD-ODA I, CD-ODA II, and 2D-ODA).

Example 1

In this example, we assume that (n − 1)∕4 nodes are interested in each topic enumerated as {10, 20}, {20, 30},  {30, 40}, and {40, 50} (Fig. 1). In addition, node u has subscribed to topics {10, 20, 30, 40, 50, 60}.

Figure 1 Implementation of the CD-MAX algorithm over Example 1.

According to the CD-ODA algorithms, node u is the center of all topics, and it will be connected to all the other nodes. Figure 1 shows that the CD-MAX algorithm constructs an overlay with a maximum node degree of (2∗(n − 1)∕4). In this example, the CD-MAX algorithm improved the maximum degree of the overlay by 50%. In this case, the CD-MAX-Ref did not improve the node degree.

Example 2

In this example, it is assumed to have four different collections interested in four different topics as listed in Table 1. In order to construct the overlay network all CD algorithms (including CD-MAX) need 6n-2 edges. The overlay network could be generated by the CD-MAX algorithm at a maximum node degree at a rate of 3n-1, whereas previous constant diameter algorithms have the maximum node degree at a rate of 4n-1.

Table 1 Example 2— topic assignments.

Nodes	Topics	
Collection A	10,20	
Collection B	10,30	
Collection C	20,30	
Collection D	10,40	

The red and black arrows demonstrate the implementation of CD-MAX and other existing algorithms respectively (Fig. 2). This case is not valid for CD-MAX-Ref, which does not have a lower maximum node degree for this example.

Figure 2 Implementation of CD-MAX algorithm over Example 2.

Evaluation of CD-MAX-Ref with Examples

Example 3

Table 2 presents eight different nodes located throughout the network. The nodes with their respective topics deployed over the network and their degrees are shown. Node number 7 has the lowest node degree; hence, it is selected as the first node to be connected to its interest group. The node becomes the center of topics 1,6. Therefore, topics 1 and 6 are removed from the original topic list, and the topics list after step 1 becomes {0, 2, 3, 4, 5, 7, 8, 9}.

Table 2 Example 3—topic assignments.

Nodes	Topics	Degree	
0	{1,2,3,5,7,8}	7	
1	{0,1,3,5,7,8,9}	6	
2	{1,4,5}	6	
3	{2,4,6}	6	
4	{0,2,3,4,9}	6	
5	{2,3,6}	6	
6	{2,5}	6	
7	{1,6}	5	

As previously explained, when more than two nodes have an equal lowest node degree, the node with the higher node density will be selected as a topic center. In the second step, nodes 1 to 6 have the lowest node degrees. However, node 1 has the highest node density (Fig. 3); hence, this node is selected to be connected to its interest group (Fig. 4). As a result, node 1 becomes the center of topics {0, 1, 3, 5, 7, 8, 9}. The remaining topic list will be {2, 4}.

Figure 3 Implementation of the CD-MAX algorithm over Example 3 Part 1.

Figure 4 Implementation of the CD-MAX algorithm over Example 3 Part 2.

In the following steps, as shown in Figs. 5 and 6, nodes 2 and 6 are selected to become the center of topics 4 and 2, respectively. Consequently, topics 4 and 2 are removed from the topic list.

Figure 5 Implementation of the CD-MAX algorithm over Example 3 Part 3.

Figure 6 Implementation of the CD-MAX algorithm over Example 3 Part 4.

Figure 6 demonstrates the result of the CD-MAX implementation over Example 3. For this overlay, CD-MAX provides an overlay with a maximum node degree of 6. The CD-MAX-Ref algorithm is applied to the resulting overlay to further reduce the node degree. The node with the highest degree (i.e. node 1) is checked, and CD-MAX-Ref finds other nodes, which are interested in topics {0, 3, 5, 7, 8, 9}, and have lower node degrees. Therefore, all the edges connecting node 1 to its interest group are removed. Nodes 0 and 4 are then selected to be the center of topic sets {5, 7, 8} and {0, 3, 9}, respectively (Fig. 7). As visualized in Fig. 3, since node 7 has the lowest node degree, it is chosen as the first node to be connected to its interest group and it becomes the center of topics 1,6.

Figure 7 Implementation of CD-MAX-Ref algorithm over Example 3 Part 5.

As visualized in Fig. 4, Nodes 1 to 6 have the lowest node degrees. But, node 1 is the node which has the highest node density; hence, this node is selected to be connected to its interest group and it becomes the center of topics 0,1,3,5,7,8,9.

The next node to be considered is node number 7, which maintains the maximum node degree of the overlay at 5. As visualized in Fig. 5, Node 2 is the best option as it has the lowest node degree and it becomes the center of topic 4. After that, in the next step, Node 6 is the best option as it has the lowest node degree and it becomes the center of topic 2 (see Fig. 6).

Therefore, CD-MAX-Ref should find alternative nodes for topics 1 and 6. All the edges joined to node number 7 are removed as node numbers 1 and 5 become the center of topics 1 and 6, respectively (Fig. 8). Consequently, the maximum node degree of the overlay decreases by 2.

Figure 8 Implementation of CD-MAX-Ref algorithm over Example 3 Part 6.

Example 4

In this example, there are (3n∕2) + 1 nodes placed over the network. Each node subscribes to specific topics which are listed in Table 3. In order to construct the overlay network, all three existing CD algorithms (CD-ODA, CD-ODA I, CD-ODA II, and 2D-ODA), the node interested in topics {x1, x2, x3, …, xn} acts as a center of the overlay which would be connected to all other nodes participating in the network.

However, the CD-MAX functions in a different manner. For example, the node which subscribes to x1 is connected to the nodes which are interested in x1,2 and {x1, x2, x3, …, xn}. This algorithm provides the overlay with a maximum node degree n related to the node interested in topics {x1, x2, x3, …, xn}. After this step, CD-MAX-Ref re-constructs the overlay with a lower maximum node degree provided by the normal CD-MAX. To illustrate, the node that is interested in {x1, x2, x3, …, xn} has the maximum node degree. All edges which are connected to this node are removed (Grey Arrows) and CD-MAX-Ref then finds other nodes with lower node degree (Fig. 9). These new nodes (N(i,j)) take over the responsibility of the topics, which are already subscribed by node u. Note that nodes N(i,j) have priority over nodes Ni and Nj through the CD-MAX-Ref implementation. The red arrows imply the new edges that are added by the CD-MAX-Ref algorithm. In this example, CD-MAX-Ref plays a key role in decreasing the maximum node degree down to (n)∕2.

Table 3 Example 4—topic assignments.

Nodes	Topics	
Ni	{xi}	
N(i,j)	{x(i,j)}	
N(1,2,3,…,n)	{x(1,2,3,…,n)}	

Figure 9 Implementation of CD-MAX algorithm over Example 4.

As a summary of all examples, in Table 4, maximum node degrees for overlay networks are listed for the existing CD algorithms in comparison with the CD-MAX algorithms.

Table 4 Maximum node degree of overlay networks built by CD algorithms.

	Examples	
Algorithm name	I	II	III	IV	
CD-ODA	n-1	4n-1	8n-1	3n/2	
CD-ODA I	n-1	4n-1	8n-1	3n/2	
CD-ODA II	n-1	4n-1	7n-1	3n/2	
2D-ODA	n-1	4n-1	7n-1	3n/2	
CD-MAX & CD-MAX-Ref	(2∗(n − 1)∕4) + 1	3n − 1	5n − 1	n∕2	

Experimental Results

This section presents a comparative evaluation of our proposed method using different overlay networks. The algorithm comparisons were conducted based on the average and maximum node degree over the resulting overlays. Both the number of topics and the number of nodes varied throughout the simulation. As noted earlier, each node has a specific subscription size and subscribes to 10 topics because of the memory restriction in the experiments. Note that only in the last simulation, this number was increased to values between 15 and 35. Each node n ∈ N can be interested in each topic t ∈ T with a probability of pi, in which ∑ipi = 1. The topic distribution probabilities pi have a Zipf distribution with α = 0.5 as used in similar studies by Carvalho, Araujo & Rodrigues (2005) and Liu, Ramasubramanian & Sirer (2005). During the experiments, we considered the impact of varying the number of nodes, topics, and topic popularity distribution on the average and maximum node degrees.

Throughout the experiments, we presented the results of the CD-ODA II algorithm as the representative of the CD-ODA algorithm series because of its better performance. The Tables 5–12 have confidence values added to the tables. Since the values shown are the average of the multiple executions with smooth differences, the confidence intervals are calculated. The calculated confidence intervals are also depicted in the aforementioned tables. The confidence interval of the results are calculated as: confidenceα,std_dev,n=norms_inv1−α∕2.std_dev∕n

where:

Table 5 Average node degree for different number of nodes (mean degree with confidence α = 0.05).

	GM	CD-ODA-II	2D-ODA	CD-MAX	CD-MAX-Ref	
Nodes	Degree	Conf.	Degree	Conf.	Degree	Conf.	Degree	Conf.	Degree	Conf.	
200	6,62	±0,05	13,56	±0,07	14,41	±0,11	13,92	±0,11	18,26	±0,09	
300	6,25	±0,03	13,79	±0,07	14,73	±0,11	14,07	±0,08	18,97	±0,07	
400	5,96	±0,03	13,83	±0,05	14,50	±0,14	14,17	±0,09	19,26	±0,02	
500	5,80	±0,02	13,91	±0,07	14,67	±0,09	14,19	±0,07	19,43	±0,01	
750	5,52	±0,01	13,98	±0,03	14,60	±0,11	14,23	±0,07	19,62	±0,01	
1000	5,38	±0,01	13,97	±0,05	14,50	±0,11	14,27	±0,05	19,72	±0,01	
2500	4,94	±0,01	14,02	±0,02	14,37	±0,08	14,31	±0,04	19,89	±0,00	
5000	4,93	±0,02	14,01	±0,03	14,40	±0,10	14,37	±0,06	19,89	±0,02	

Table 6 Maximum node degree for different number of nodes (mean degree with confidence α = 0.05).

	GM	CD-ODA-II	2D-ODA	CD-MAX	CD-MAX-Ref	
Nodes	Degree	Conf.	Degree	Conf.	Degree	Conf.	Degree	Conf.	Degree	Conf.	
200	19,80	±1,30	137,80	±2,77	131,40	±1,94	113,70	±1,98	38,10	±0,90	
300	21,50	±0,00	203,80	±3,68	199,10	±3,26	176,80	±1,62	51,30	±1,34	
400	22,80	±0,00	272,60	±2,45	261,40	±4,93	237,50	±1,99	62,60	±1,44	
500	22,20	±0,00	337,00	±1,98	334,70	±6,27	303,40	±2,13	75,20	±1,84	
750	22,60	±0,00	510,90	±3,72	498,30	±5,00	458,70	±3,71	102,90	±2,87	
1000	23,90	±0,00	678,60	±4,79	677,30	±6,91	619,10	±3,41	132,60	±3,11	
2500	32,60	±0,00	1686,70	±7,61	1683,60	±7,10	1584,20	±6,78	296,67	±2,36	
5000	36,00	±0,00	1692,00	±7,12	1685,00	±9,08	1580,00	±7,65	296,00	±3,08	

Table 7 Average node degree for different number of topics (mean degree with confidence α = 0.05).

	GM	CD-ODA-II	2D-ODA	CD-MAX	CD-MAX-Ref	
Topics	Degree	Conf.	Degree	Conf.	Degree	Conf.	Degree	Conf.	Degree	Conf.	
200	9,99	±0,11	13,16	±0,07	12,80	±0,07	13,53	±0,12	14,52	±0,11	
250	10,38	±0,11	12,67	±0,05	12,21	±0,04	13,06	±0,12	13,74	±0,12	
300	10,67	±0,12	12,27	±0,11	11,78	±0,09	12,63	±0,08	12,99	±0,12	
350	10,65	±0,06	11,77	±0,06	11,36	±0,05	12,14	±0,08	12,34	±0,07	
400	10,37	±0,04	11,25	±0,04	10,88	±0,06	11,59	±0,09	11,81	±0,09	

norms_inv = Inverse of the standard normal cumulative distribution with the given probability

std_dev = Standard deviation of the given values

n = number of values (in this study, number of runs, i.e., 10)

Table 8 Maximum node degree for different number of topics (mean degree with confidence α = 0.05).

	GM	CD-ODA-II	2D-ODA	CD-MAX	CD-MAX-Ref	
Topics	Degree	Conf.	Degree	Conf.	Degree	Conf.	Degree	Conf.	Degree	Conf.	
200	22,10	±1,29	42,90	±1,22	30,30	±3,43	29,90	±0,85	18,70	±0,51	
250	21,90	±1,22	37,00	±1,78	26,90	±1,95	23,80	±0,49	16,90	±0,46	
300	22,70	±1,21	31,70	±1,13	23,80	±1,54	20,10	±0,46	15,70	±0,51	
350	23,00	±0,83	28,00	±1,49	22,60	±0,98	17,50	±0,33	15,40	±0,43	
400	20,40	±1,25	26,10	±1,66	21,60	±2,03	15,70	±0,30	14,30	±0,30	

Table 9 Average node degree for different subscription size (mean degree with confidence α = 0.05).

	GM	CD-ODA-II	2D-ODA	CD-MAX	CD-MAX-Ref	
Subsc.	Degree	Conf.	Degree	Conf.	Degree	Conf.	Degree	Conf.	Degree	Conf.	
15	12,86	±0,07	29,92	±0,16	32,28	±0,43	15,09	±0,12	27,93	±0,07	
20	12,20	±0,06	28,21	±0,19	29,42	±0,79	14,19	±0,14	37,08	±0,10	
25	11,51	±0,06	25,34	±0,41	26,31	±0,44	12,62	±0,23	46,04	±0,10	
30	10,83	±0,03	22,31	±0,38	23,08	±0,43	11,20	±0,23	11,20	±0,23	
35	10,21	±0,04	20,93	±0,32	20,90	±0,48	10,46	±0,16	10,46	±0,16	

Table 10 Maximum node degree for different subscription size (mean degree with confidence α = 0.05).

	GM	CD-ODA-II	2D-ODA	CD-MAX	CD-MAX-Ref	
Subsc.	Degree	Conf.	Degree	Conf.	Degree	Conf.	Degree	Conf.	Degree	Conf.	
15	16,10	±0,81	185,80	±1,38	183,90	±1,97	174,10	±0,85	51,90	±0,81	
20	14,10	±0,94	198,00	±0,48	197,60	±0,50	193,60	±0,50	68,70	±1,80	
25	13,00	±1,49	199,00	±0,00	198,90	±0,19	197,90	±0,19	82,20	±0,91	
30	10,60	±0,97	199,00	±0,00	199,00	±0,00	198,80	±0,25	198,80	±0,25	
35	9,40	±0,84	199,00	±0,00	199,00	±0,00	199,00	±0,00	199,00	±0,00	

Table 11 Running time for different number of nodes in seconds (mean values with conf. α = 0.05).

	GM	CD-ODA-II	2D-ODA	CD-MAX	CD-MAX-Ref	
Nodes	Avg.	Conf.	Avg.	Conf.	Avg.	Conf.	Avg.	Conf.	Avg.	Conf.	
200	0,12	±0,01	0,29	±0,03	0,11	±0,01	0,29	±0,02	9,72	±0,38	
300	0,38	±0,02	0,61	±0,04	0,26	±0,02	0,64	±0,03	31,13	±0,95	
400	0,88	±0,06	1,06	±0,08	0,42	±0,02	1,12	±0,05	68,12	±2,08	
500	2,09	±0,83	1,69	±0,27	0,63	±0,04	1,62	±0,05	122,49	±1,55	
750	5,52	±0,17	3,44	±0,08	1,47	±0,07	3,50	±0,10	387,86	±4,48	
1000	12,08	±0,04	5,67	±0,12	2,56	±0,05	5,84	±0,15	922,76	±25,67	
2500	211,91	±5,28	37,95	±1,42	19,69	±1,61	38,90	±2,26	4856,00	±150,14	
5000	262,86	±2,23	46,10	±1,25	23,56	±0,95	50,20	±2,45	15049,50	±402,14	

Table 12 Running time for different subs. size in seconds (mean values with confidence α = 0.05).

	GM	CD-ODA-II	2D-ODA	CD-MAX	CD-MAX-Ref	
Subsc.	Avg.	Conf.	Avg.	Conf.	Avg.	Conf.	Avg.	Conf.	Avg.	Conf.	
15	0,16	±0,00	0,20	±0,01	0,15	±0,00	0,19	±0,01	9,32	±0,41	
20	0,22	±0,01	0,15	±0,01	0,22	±0,00	0,15	±0,01	9,18	±0,31	
25	0,28	±0,01	0,14	±0,01	0,31	±0,01	0,13	±0,01	9,68	±2,45	
30	0,35	±0,01	0,13	±0,01	0,40	±0,01	0,12	±0,01	14,61	±2,12	
35	0,43	±0,01	0,12	±0,00	0,52	±0,01	0,11	±0,01	15,39	±1,77	

α = significance level, which is a probability between 0 and 1.

Average and maximum node degree values for varying number of nodes

In this experiment, the number of nodes varied between 200 to 5000, while the number of topics was kept constant (100). The subscription size was fixed at 10. Each node randomly subscribed to different topics. The average node degrees (see Table 5 and Fig. 10) provided by each algorithm slightly decreased as the number of nodes increased. This result indicates that because of the increased probability of having overlaid edges in sub-graphs, a smaller number of edges connected a larger number of nodes, and the average node degree of the overlay will decrease. However, the maximum node degree of the constant-diameter algorithms increased with the increasing number of nodes (see Table 6 and Fig. 11). This observation is also valid in the case of the 2D-ODA algorithm.

Figure 10 Average node degree for different number of nodes.

Figure 11 Maximum node degree for different number of nodes.

Unlike the other constant-diameter algorithms, in which a small number of nodes covered most of the topics, to decrease the node degree of the overlay, the CD-MAX and CD-MAX-Ref algorithms chose nodes with a lower correlation to become the center of the topics. More edges would be needed to connect the nodes, thereby raising the average node degree.

Compared with the GM algorithm, CD-MAX and CD-MAX-Ref for 1000 nodes requires 2,65 and 3,66 times more edges in average, respectively (see Table 5). Considering the maximum node degree with a growing number of nodes, more nodes should be connected to the overlay center nodes. Thus, the maximum node degree provided by every constant-diameter algorithm sharply increased (see Table 6). The maximum node degree of the GM algorithm will decrease relatively (does not increase while others are increasing fast enough) because more nodes with a higher correlation distributed the node degree of the overlay. Although the GM algorithm had low maximum and average node degree, it had a higher diameter Chockler et al. (2007a), Onus & Richa (2011) and Carvalho, Araujo & Rodrigues (2005). While the CD-MAX required less number of edges to build the overlay network compared to other ODA algorithms, CD-MAX-Ref outperformed all competitors by at least four times. In addition, it became the only algorithm with the ability of approaching the performance of the GM algorithm.

Average and maximum node degree values for varying number of topics

In this experiment, the number of nodes and the subscription size were fixed at 100 and 10, respectively. Meanwhile, the number of topics varied from 200 to 400. The overlays face two different conditions when the number of topics was increased. First, the correlation between the nodes decreased; thus, more edges were used to connect the nodes. Second, the number of nodes without any connection to a neighbor increased. The average node degree will increase if the first condition dominates. The entry in Table 7, which corresponds to CD-MAX-Ref with a number of topics of 250, is an indication of the first condition. In contrast, the average node degree will decrease if the second condition dominates. The second condition has a greater effect than the first condition (see Table 7 and Fig. 12). Hence, the overall average node degree of the overlay for every algorithm reduced when the quantity of topics increased.

Figure 12 Average node degree for different number of topics.

The second condition affected the maximum degree of all the algorithms. Table 8 and Fig. 13 shows that the maximum node degree of all the algorithms decreased as the set of topics indicated more diversity.

Figure 13 Maximum node degree for different number of topics.

Average and maximum node degree values for varying subscription size

For the final experiment, the numbers of nodes and topics were kept at 200 and 100, respectively. However, the subscription size was varied between 15 and 35. As noted earlier, each node randomly subscribed to different topics using the interest function. Tables 9–10 and Figs. 14–15 illustrate the effects of changing the subscription size on the selected algorithms. As the subscription size grows, the nodes can get connected with each other with a higher correlation rate. Subsequently, the rate of the average node degree decreases. Meanwhile, the contribution between the nodes rises when the subscription size grows. Therefore, the GM algorithm can find many node pairs, which dramatically reduces the total number of topic-connected components. Hence, the maximum node degree will decrease as the subscription size increases. For all the algorithms with a star topology, the maximum node degree will increase as a single node may be selected as the center of many topics. The results of the final set of experiments showed that CD-MAX required slightly lesser number of edges to build the overlay network when compared with the other algorithms. From the node degree perspective, all constant-diameter algorithms, including the proposed algorithms, showed similar averages. The decrease in the average node degree in the CD-MAX family of algorithms was slightly higher than that in the other competitors, resulting in a higher scalability.

Figure 14 Average node degree for different number of subscriptions.

In this experiment, CD-MAX-Ref algorithm tries to find a node with the minimum number of neighbors. In other words, a node with the least connection with others is always an ideal one to be selected. As it is assumed that this algorithm tries to keep the maximum node degree low. Hence, this trend makes the average node degree higher and decrease the maximum node degree of the overlay as the subscription size goes up (see Fig. 14). However, as the subscription size increases, the overlay reaches to a threshold at which it is almost impossible to find a node with a lower number of neighbors. Subscription size has been increased and all nodes have higher correlation with each other. Inevitably, the average node degree sharply decreases and the maximum node degree rises more than expected (See Fig. 15).

Figure 15 Maximum node degree for different number of subscriptions.

Comparison of the running time cost of the algorithms

An optimized overlay network not only can forward packets with shorter latencies, but also improve the maintenance of connections and provide resiliency against link failures Besta & Hoefler (2014). This optimization is closely dependent on the total degree of nodes. Although designing a topology for optimum resource consumption is not possible, minimizing the total number of links and network diameter and optimizing the algorithm in terms of the time complexity can be considered as distinguishing metrics Voulgaris et al. (2005) and Chockler et al. (2007b). Therefore, we included an analytic and experimental run-time analysis of the algorithms for comparison. Tables 11 and 12 include the running times of all the evaluated algorithms and Figs. 16 and 17 illustrates the running times in logarithmic scale. The running time of the CD-MAX-Ref algorithm was higher than those of all the other algorithms. Table 11 shows that the growth rate of the running time for the ODA algorithms are similar while the nodes were increasing from 200 to 5000. The growth rate for CD-MAX-Ref and GM are similar, while that for the CD-MAX algorithm is lower. These promising results of CD-MAX illustrate its suitability for the number of nodes that increases beyond a threshold. CD-MAX is the fastest algorithm for building and maintaining the requested topology. Meanwhile, Table 12 shows that the running time for CD-MAX and CD-ODA decreased as the number of subscriptions increased. Considering this, CD-MAX is the fastest algorithm beyond a subscription size of 25. Table 12 indicates that CD-MAX is the fastest algorithm among all algorithms, including the GM algorithm. The relatively higher speed of CD-MAX makes it a more suitable option for dynamic environments, where arrivals and departures are higher.

Figure 16 Running time for different number of nodes in seconds.

Figure 17 Running time for different subscription size in seconds.

The experiment was carried out under Windows operating system using a computer with an I7-7700HQ processor. As noted in Tables 11 and 12, the confidence intervals may fluctuate among the number of nodes and subscriptions. Due to operating system processor loads in some runs, the running times resulted in higher confidence intervals.

Conclusions and Future Work

This study presented a novel algorithm (i.e., CD-MAX) that provides overlay networks with a minimum node degree and a low diameter of 2. The algorithm was implemented in a decentralized manner, and was fast and scalable. The proposed algorithm considerably decreased the maximum node degree, thereby resulting in an overlay network that was more scalable compared to the other algorithms studied. The minimization of the maximum degree plays a key role in a number of networks of very large domains, such as survivable and wireless networks.

The study results indicated that the proposed algorithm outperforms the ODA-based algorithms in terms of the decreased diameter and average node degree of the overlay networks and approaches the performance of the GM algorithm.

Our analytic and experimental results revealed that the running time of the proposed CD-MAX algorithm is similar to CD-ODA-II and 2D-ODA on average node degree, while outperforming all ODA algorithms excluding GM algorithm on maximum node degree (see Tables 5–6 and Figs. 10-11). In addition, CD-MAX builds the network in similar time with ODA algorithms with better maximum node degrees, whereas GM fails to scale out with higher node counts (see Table 11 and Fig. 16). As the number of topics increases, the average node degree for CD-MAX and others are similar; however, the CD-MAX and CD-MAX-Ref outperforms on maximum node degree. (See Table 7–8 and Figs. 12–13).In terms of different subscription sizes, the CD-MAX algorithm outperforms other ODA algorithms excluding GM on average node degree (see Table 9).

By combining the results obtained from the running time experiments and the measurements of the maximum node degree, we can assert that the CD-MAX algorithm is more suitable for networks requiring high scalability because it simultaneously reduces the communication costs and the running time. In contrast, the CD-MAX-Ref algorithm best suits environments with slow and gradual changes (i.e., having a low churn rate) and those with a large number of topics, and subscription sizes, which are characteristics of typical internet-based systems.

As a future work, studies may concentrate on the simulation of millions of nodes, topics, and subscriptions with a typical churn rate of internet users by employing big data and high-performance computing environments. Further research should consider achieving the best average and low maximum node degrees while optimizing the running time. The proposed algorithms can further be extended to include power consumption optimization (Alsultan, Oztoprak & Hassanpour, 2016), and location awareness to build clustered overlay structures to reduce delay, increase bandwidth, improve scalability (Bozdagi & Oztoprak, 2008), resilience to node failures, and have load balancing.

Supplemental Information

Supplemental Information 1 Experimental studies for all compared algorithms and the implementation for the proposed algorithms

Click here for additional data file.

Additional Information and Declarations

Competing Interests

Author Contributions

Data Availability

The authors declare there are no competing interests.

Semih Yumusak conceived and designed the experiments, performed the experiments, analyzed the data, performed the computation work, prepared figures and/or tables, authored or reviewed drafts of the paper, and approved the final draft.

Sina Layazali conceived and designed the experiments, performed the experiments, performed the computation work, prepared figures and/or tables, authored or reviewed drafts of the paper, and approved the final draft.

Kasim Oztoprak and Reza Hassanpour analyzed the data, authored or reviewed drafts of the paper, and approved the final draft.

The following information was supplied regarding data availability:

All algorithms used in the experiments (GM, CD-ODA II, 2D-ODA, CD-MAX, CD-MAX-Ref) are available in the Supplementary File. The execution of the source code provides the raw data for the analysis of our proposed algorithms CD-MAX and CD-MAX-Ref.

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
