# Peer review of "Low-diameter topic-based pub/sub overlay network construction with minimum–maximum node degree"

_PeerJ Computer Science, doi:10.7717/peerj-cs.538_

## Round 0.1 · original submission · Major Revisions

The reviewers have identified a couple of shortcomings, and therefore I recommend a major revision of this submission. It will be necessary to increase the node-count in the experiments to numbers that compare to related studies, the statistical significance has to be reported and we need high confidence in the results. The algorithm should be better described and a complexity analysis could be useful. Please also address all issues that have been mentioned with regard to the reporting.

Reviewer 1 ·

Basic reporting

Overall, your paper is written in a clear and understandable fashion.
However, there are multiple instances of broken math notation (e.g., lines 157, 164, 260, 261, and table 3).

The introduction could benefit from example usecases of publish/subscribe networks.
In lines 45-54, you should add a concrete explanation on how messages actually travel from publisher to subscribers using the sub graph.
It is not clear to me why these particular systems form lines 79 to 99 where chosen as background on publish/subscribe systems.
Lines 113 to 120 use notation that is only introduced later.

Your pseudocode and algorithm descriptions are clear and understandable.
Your figures would benefit from more information, both as part of the figure and in the caption.
As is, the steps explained in figures 2-7 are only understandable with parallel examination of the corresponding sections of your text.
To improve this, add a short description of the executed step to each caption and augment the figures themselves with information such as the current list of topics T.
Further, Figure 1 contains an error:
Node u's topic list is missing topic 60, which u has subscribed to according to line 194.
While your examples are helpful in understanding how your algorithms work, the wording of the example's titles makes them sound like contradicting statements.

As required, your paper is self-contained:
You state your research question, present your proposed algorithms and evaluate them to show that they meet their expectations.

The preliminary section mostly defines the required terms in an understandable way.
It might not hurt to state that |x| denotes the number of elements in x.
Further, there is no definition of the diameter of an overlay network, despite it being a central term of your paper.

Experimental design

Lines 55 to 71 clearly define the relevance of your research question and how your proposed algorithms will improve the current state of the art.

Related work (Chockler 2007a, Section 8), seems to use a much greater number of nodes (up to 10,000 compared to your 100) in their evaluation.
Could you please explain why you you have not evaluated your results using a greater number of nodes.

You might be able to improve your evaluation by proving bounds of your algorithms, similar to lemmas 6.5 and 6.6. in Chockler 2007a.

Validity of the findings

As noted previously, your pseudocode and description of the algorithm is clear and understandable.
However, your included source code lacks any documentation.
The output of the program does not indicate which of your experiments is run.

Line 350 could have clearer formulation, since Tables 11 and 12 do not reveal on first glance that CD-MAX is faster.
You should provide more context at which point it becomes infeasible to use CD-MAXref over CD-MAX, seeing that is has a much greater running time.
Overall, I can see the links between your conclusions and the results of your evaluation.

Additional comments

It's a bit of a stretch to call a paper form 2007 "recent" (line 54).

·

Basic reporting

The paper is well written and does a good job of motivating the problem. The authors furthermore present a clear comparison to the related work and put their contributions into context. Raw data is provided, figures are appropriate and easy to understand.

The paper is mostly well structured, though I propose to spend less time on examples and put more emphasis on proofing that (steps of) the algorithms achieve the desired properties.

The main drawback of the paper is the lack of clarity in both the algorithm description and the proof. A more minor issues are some aspects in the notation.

=====Proof for Algorithm 2======
There are two issues in the proof of Algorithm 2:
i) The algorithm eliminates all edges of a node associated with a topic, however, what happens if the edge is associated with more than one topic? it seems to me like there might be removals that lead to nodes not receiving all topic information. If I misunderstand, you have to explain in the proof why that is not the case.
ii) The pseudocode searches for a node q with d_q + d_t <d_u − d_t whereas the text (line 179, page 5) states that the algorithm finds a node with d_q + d_t <d_u...Please either clarify or fix this seeming inconsistency

=====Proof for Algorithm 1======
There is no proof for Algorithm 1 but I think it is non-obvious that the algorithm does indeed include edges for all topics and users and does not accidentally remove some in Line 18.

=====Pseudocode=====
The pseudocode is unclear on multiple occasions, all of which can probably be clarified. For Algorithm 1:
-Line 4-7: I assume you'd calculated the maximum after calculating each individual value. Maybe you mean that you compare the old maximum with the value for node u but the code seems like you compare all the values again.
-Line 8: The term 'max_n is not a single' is unclear, I assume you mean |Max_n|>1, however, it isn't even clear that Max_n is a set up to that point, so it's a bit sloppy
-Line 9: What happens if there is more than one element in Max_n with maximal degree?
For Algorithm 2:
- My interpretation of all the —s is minus but that does not make sense as the algorithm would never enter the while loop (because -\infty < -FindMaxNodeDegree()), so maybe it means something else or they should just not be there.
- what happens if there are several such nodes q (Line 10)
- I think you never really defined centre node

======Notation=====
There are some small unclarities in the mathematical notation that make reading harder and require some guessing as to what is meant:
- p.3: In Line 114-115, you define Int(x,m) as a binary function and then use it as a set in Line 119
- p.3: Line 113-114: “There are m ∈ M|n : Int(n,m) = 1| individual” This sounds like m ∈ M|n : Int(n,m) = 1| is a number but it’s not (do you mean the sum over all m?)
- p.4: Eq. 2, the function e(u,v) is not defined, it could be e=(u,v)?
- p.4: Eq. 4, can you give an intuition of what the equation conveys? You also state it’s normalized but it seems like the denominator can easily be bigger than the nominator
- p.5, Line 164, du -> d_u
- p. 11, Line 260: there seem to be some \in missing

Experimental design

The experimental design considers all relevant parameters and explores meaningful scenarios. A meaningful comparison to other algorithms is conducted. It is mainly well explained but for minor two aspects:
i) the number of runs is not given
ii) the part on the probability pi is not clear: first it is set to 1, then it is said that it follows a zipf distribution…please clarify

Validity of the findings

The conclusions presented by the authors are well justified and provide an in-depth comparison of multiple algorithms. Overall, the validation is one of the strength of the paper. The only aspect that needs to be improved is that the authors do not present statistical tests, standard deviations, or confidence intervals to show that the results are indeed statistical significant. Such an analysis has to be added to the revised version.

Additional comments

I liked the idea of the algorithms but I was wondering if there are not other criteria that play a similar important role as diameter and degree, such as i) resilience to node failures, ii) load balancing. Your algorithm might result in edges that distribute content for many topics. The limited bandwidth of these edges might delay dissemination.

Reviewer 3 ·

Basic reporting

The english is ok, literature references are ok as well. The article structure is mainly ok, but the authors also explain in length five different examples on topologies generated by their algorithm, which is too much. As a result the paper looks a bit unbalanced.
The experimental results are displayed in tables only. Graphs would probably be simpler to read.

Experimental design

See general comments.

Validity of the findings

See general comments.

Additional comments

The paper proposes a novel centralized algorithm to construct low-diameter topologies intended for publish-subscribe content distribution. This is definitely an interesting topic.

However, I am not really confident about this paper. First of all, the proposed algorithm is not presented in sufficient detail. The authors put the actual algorithm, but the algorithm itself is not explained in the text. As a result, the algorithm as main contribution is introduced on a bit more than one page. Instead the authors give in total 5 examples on topologies produced by their algorithm when some nodes and their subscriptions are given as input. That is for sure too much. When the algorithm is explained properly, one or two examples should actually be enough and the authors would also save a lot of space.
Furthermore, the paper could benefit from a bit more theoretical foundation. A complexity analysis would be interesting here. Moreover, the whole problem could be formalized as optimization problem, which probably someone in related work already did. However, this optimization problem could also serve as a benchmark on how close the proposed algorithm actually can get to the optimum (for small node numbers at least).

On the positive side, it has to be mentioned that the authors compare their approach to several approaches from related work. Nevertheless, none of these approaches seems to be contained in the related work section of the paper. I would recommend to introduce and explain them shortly.

Regarding the experimental results I am not sure if the metrics used by the authors (average and maximum node degree) give the complete picture here. What about the number of edges? What about additional metrics that quantify the load on single nodes, e.g., centrality?

Minor comments:
* * *
* page 4: you can refer to lines in the algorithm from the text to explain what it actually does
* page 5: "with a lower node degree du" -> du? is that notation or do you mean "do"?
* page 10: "all three existing CD algorithms (CD-ODA, CD-ODA, CD-ODA, and 2d-ODA)" -> that is three times the same algorithm and in total 4 instead of 3.
* Page 10,¸ Figure 10: "To illustrate, node u that is ..." -> where is node u in Figure 10?
* Page 11: what does a probability pi of 1 mean? 100% probability??? probably not. Please explain.
* Page 11: "Each node n N" -> do you mean "n \in N" (n element of N)?, same for "t T"
* Page 11: All subsection heading in the result section start with the misleading term "Fixed Average and Maximum Node Degree". I understood that your approach generates low-diameter topologies and tries to minimize the maximum node degree, but how can the maximum degree then be static? In your results you again refer to results as CD-MAX and this values differ depending on the node size.
* Page 11: What is the diameter that your algorithm tries to reach here? 2? Not mentioned.

---

## Round 0.2 · Minor Revisions

Please address the remaining requests by the reviewers, thanks!

Reviewer 1 ·

Basic reporting

The updates section require more proof-reading, e.g.:

- L45: Pub/sub system *have* a variety of use cases.
- L65: *A* constructed..
- L73:
1. What exactly do you mean by "decoupling"
2. I'm sure one can find examples where decoupling is not advantageous
3. Either "systems with decoupling mechanism*s*" or "systems with *a* decoupling mechanism"
- L152: Why not use $(u,v)\in E$ instead of introducing $e(u,v)$?
- L182: *The* selected node...
- L184: ..., *the* pub/sub overlay network
- L303: ...which *are* listed in...
- L329:
- Tables 5-12 *have* confidence values...
- ...results are calculated as**:**

Experimental design

Please give some more intuition on why CD-MAX-Ref behaves so different from anything else in Figures 14 and 15

Validity of the findings

no comment

Additional comments

Thank you for your updated manuscript and taking the time to implement my previous comments.

·

Basic reporting

The new version is greatly improved.

However, I still believe that the correctness (in the sense of all nodes receiving messages for their topics) of Algorithm 1 is not addressed properly. The main reason stated for correctness is "Unless there is a topic which is subscribed by a single node, pub/sub overlay network is connected." but you don't i) prove that it is connected, and ii) explain exactly why connected guarantees that nodes receive all messages. I don't even understand why the statement is correct, as the PubSub overlay should not be connected if e.g., there are two topics each with three distinct parties interested.
I think your algorithm achieves what you want, but I think you are not using the proper argument here.

Similarly, you start the complexity proof in Line 216 with 'Finding a node with minimum node degree takes O(|V |2 ∗ |T |)' without giving an explanation on why that is the case.

Only remaining minor issues are:
- p.2 can you add a references that PubSub is heavily used in cloud computing systems/that the named systems use PubSub
- Line 77: Not only do β -> does
- Algorithm 1: you are still using Max_n as both a set and a number...it can only be one, you probably need two variables here
- Line 10 of Algorithm 1: u⇐w∈Maxn AND w has the largest du. -> it's not clear from which set of nodes w has the largest d_u (and shouldn't it be d_w?), is it all nodes or only those in Max_n]

Experimental design

Concerns appropriately addressed.

Validity of the findings

Concerns appropriately addressed.

---

## Round 0.3 · accepted · Accept

Thank you for the revised version, which can now be published.